# Impacts of AIOT Implementation Course on the Learning Outcomes of Senior High School Students

**Chih-Cheng Tsai [1], Yuh-Min Cheng [2], Yu-Shan Tsai [3] and Shi-Jer Lou [3,*]**

[1]  Department of Industrial Technology Education, National Kaohsiung Normal University, Kaohsiung 82446, Taiwan; khccvs280@gmail.com

[2]  Department of Computer Science and Information Engineering, Shu-Te University, Kaohsiung 82445, Taiwan; cymer@stu.edu.tw

[3]  Graduate Institute of Technological and Vocational Education, National Pingtung University of Science and Technology, Neipu 91201, Taiwan; perconle@gmail.com

*  Correspondence: lou@mail.npust.edu.tw

**Abstract:** In this study, experimental teaching was conducted through the artificial intelligence of things (AIOT) practical course, and the 4D (discover, define, develop, deliver) double diamond shape was used to design the course and plan the teaching content to observe the students' self-efficacy and learning anxiety. The technology acceptance model (TAM) concept was used to explore learning effectiveness and satisfaction. A total of 36 Senior One students from a senior high school in Kaohsiung were selected as the research subjects in two classes per week for 13 weeks. Questionnaires and qualitative interviews were used to understand the correlation between students' self-efficacy, anxiety, outcomes, and learning satisfaction with the AIOT course during the flipped learning process. The study used SmartPLS3 to analyze the measurement model and structural model, and bootstrapping to conduct the path analysis and validate the research hypotheses. Because this research provides students with relevant online teaching videos for linking and browsing in the textbooks, and the video time should not be too long, in the teaching process, small-scale online learning courses are adopted for flipped teaching, which promptly arouses students' interest and enhances their learning participation. Through the sharing of homework with each other, its content deficiencies were modified, students' anxiety in learning was reduced, and the effect of learning and thinking together was achieved; in the teaching mode, theoretical content was combined with physical and practical courses to implement cross-disciplinary. To learn, the principle of 4D double diamond design to make "reciprocal corrections" between curriculum planning and teaching implementation as the teaching model framework was used. The results of this study show "self-efficacy" has a significant positive effect on the "perceived usefulness" and "perceived ease of use" of users. "Learning anxiety" does not significantly predict the "perceived ease of use" or "perceived usefulness" of flipped learning using online e-learning. The "perceived ease of use" and "perceived usefulness" have positive impacts on the prediction of "behavioral intention" in flipped teaching using online digital teaching materials. The "perceived ease of use" has a positive and high impact on "perceived usefulness". "Learning engagement" positively affects students' actual "behavioral intention" towards learning. Students are highly receptive to learning new knowledge about science and technology.

**Keywords:** AIOT; implementation course; learning outcomes; education reform

## 1. Introduction

The purpose of this study is to investigate the development of the artificial intelligence of things (AIOT) teaching model based on a small private online course (SPOC), the curriculum design model and teaching strategies, and to propose a "SPOC–AIOT educational curriculum design model" for technical high school level. Based on this model, we

developed SPOC–AIOT teaching modules for technical high school level, and confirmed the suitability of the teaching modules through teaching experiments.

*1.1. Research Background and Motives*

While traditional classroom teaching implements "hands-on" courses by "thinking", the "maker movement" emphasizes "learning by doing" in social environments through digital applications and tools, which has completely reversed the traditional learning mode by taking "innovative thinking" as the subject of the course. This kind of learning inspires teachers, is regarded as the driving force of education [1], and has become an important research field. Maker education is closely related to science, technology, engineering, and mathematics (STEM) learning, which teaches learners to present their ideas in a practical manner according to the concept of "learning by doing", in order to cultivate students' innovative spirit and practical ability. Maker education is based on "problem oriented" and "topic oriented" learning methods and solves real problems through a collaborative learning experience [2]. Ref [3] pointed out that, while most STEM education aims to enhance creativity, solve problems, and develop new ideas and approaches, there are few skilled learning outcomes, which suggests that STEM should focus on learning outcomes related to thinking ability.

Education reform in various countries must cultivate students to be innovative talents in the 21st century to meet the era of artificial intelligence. In order to enhance national competitiveness, the deep learning of artificial intelligence is included in the course, and such educational reform also includes computational thinking, the Internet of Things (IOT), and artificial intelligence (AI) into the course outline, in order to establish students' information literacy and application ability, and cultivate their abilities of creative thinking and problem-solving through the learning process of information technology. The United States, the United Kingdom, Germany, Australia, and other countries have recognized that "computational thinking" is of great help in learning and employment and have listed information education as the priority of educational reform [4]. With the advent of industry 4.0, the IOT and AI have become the prominent subjects of emerging technologies. However, it is pointed out that, in the past, course planning for senior high schools was not based on students' needs or industrial trends, but on teachers' expertise or teaching intention, which led to significant gaps between students' professional knowledge and the industrial environment, meaning that students were unable to apply what they had learned, and it was necessary to retrain the industrial labor force [5].

How to effectively implement AIOT courses in a short time, in order that students can obtain the maximum learning outcomes, is a test of the teachers' course design and implementation. In the face of the rapid development of science and technology, learning is no longer traditional classroom learning. Massive open online courses (MOOCs), as launched in 2008 by Canadian academics, Bryan Alexander and Dave, took advantage of the internet for online learning, and since then, the courses have spread like wildfire [6]. Ref [7] conducted 60 studies on maker activities, with a high proportion (48 studies) related to the perception of learning outcomes, and twenty-eight of the subprojects were related to STEM content knowledge, which shows the importance of perceived learning results in maker activities. It can be seen that "maker learning" with students and independent learning as the core has become the new trend of learning.

In order to implement "computational thinking" and the "maker spirit" in the field of science and technology, this research conducted an AIOT course study, which mainly focused on learners and carried out course planning and teaching focus by means of the "4D double diamond model (discover, define, develop, deliver)" in the process of divergence and convergence. This study implemented the learning transfer of computational thinking from Turtle Graphics to Python practice, and then, practiced students' learning and ability of "do, use, and think" from the production and application of physical objects. After the experiment, quantitative and qualitative researches of "SPOC–AIOT (small private online course–artificial intelligence of things) implementation course" were carried out to verify

the learning outcomes. This study applied flipped teaching in the characteristic course of "AI deep learning", and then, the teaching model of the AIOT course was developed to be applicable to senior high school AI and other related courses.

### 1.2. Research Questions

This study used flipped teaching as the teaching strategy to improve the learning outcomes of students learning in the artificial intelligence (AI) deep learning courses. In terms of teaching, the researchers used the existing online AI, deep learning, Python programming language, and other relevant teaching videos to conduct 13 weeks (a total of 26 classes) of experimental teaching, in order to understand the learning outcomes and course satisfaction of flipped teaching for technology senior high school students in AI deep learning. The questions to be discussed in this study are, as follows:

1. What is the impact of "individual difference factors" on the technology acceptance model of senior high school AI deep learning?
2. What is the relationship between "learning engagement" and the technology acceptance model of senior high school AI deep learning?
3. What is the impact of the "technology acceptance model" on the learning outcomes and course satisfaction of senior high school AI deep learning?
4. How to construct a teaching module suitable for senior high school students to learn AI deep learning courses?

### 1.3. Research Purposes

1. To explore the impact of "individual difference factors" on the technology acceptance model of senior high school AI deep learning.
2. To explore the relationship between "learning engagement" and the technology acceptance model of senior high school AI deep learning.
3. To explore the impact of the "technology acceptance model" on the learning outcomes and course satisfaction of senior high school AI deep learning.
4. To construct a teaching module suitable for implementing AIOT courses in senior high school.

## 2. Literature Review

This study applies flipped teaching to the "deep learning with artificial intelligence" course and uses the "4D double diamond model" to teach from turtle drawing to Python learning, and then combines the production and application of real objects to develop students' innovation and practical skills. This research paper will explore the educational applications of 4D double diamond model, artificial intelligence, Python, deep learning, and the technology acceptance model.

### 2.1. Application of the 4D Double Diamond Model (Discover, Define, Develop, Deliver) in Education

The double diamond model, which is a famous and popular design process, is a design process model developed by the British Design Council in 2005, and is divided into the four stages of exploration, definition, development, and implementation. Its characteristics emphasize "divergent views" and "convergent thinking", which first produces many ideas, and then, refines and reduces them to the best ideas. This model includes "divergent views" and "convergent thinking", one is to confirm the definition of the problem, and the other is to create a solution. The double diamond model shows four stages between two adjacent diamonds, each of which is characterized by fusion or divergent thinking.

- Exploration of the problem in stage 1: to understand the real world behavior and the problems faced.
- Definition of the problem in stage 2: after understanding the actual usage environment and the problems encountered, record and prioritize the problem to "determine the problem to be solved".

- Development of possible solutions in stage 3: to evaluate solutions to determine the best implementation.
- Selection and development of the solution in stage 4: to finalize the solution.

AI is becoming more and more widely used in the design process. In order to promote AI technology to effectively assist in design and meet the design requirements of professional designers, [8] the "double diamond design model" was proposed to provide infinite possibilities for future design tools, which starts from the expansion stage of students' thinking, discusses the role that AI should play in the design process, and proposes ideas for future design tools. In response to the diversity of students, [9] a cross-platform service blueprint was designed with the "double diamond design flow model" to solve various challenges for students in universities. In this study, technology senior high school AIOT characteristic courses were implemented by adopting the 4D double diamond model (discover, define, develop, deliver) as the framework of the course and teaching design, which is intended to guide students' learning in the four stages of exploration, definition, development, and implementation, and is developed from "implementation" to adapt to the current trend of AIOT, as applied in daily life.

### 2.2. Impact and Application of AI in Education

AI can be traced back to the Turing machine, designed and manufactured by Alan Turing in 1936, which is the most important prototype of modern computer logic in computer science [10]. In fact, AI is not a new science, but with the progress of computer software and hardware, it has become an emerging technology easily implemented in daily life. As a result, countries around the world are developing plans to address the appropriate use of AI and determine its wide impacts on training, lifelong learning, and education. Research shows that an increasing number of innovative tools, smart applications, and methods are transforming education systems to improve the learner experience and classroom learning outcomes [11,12]. Ref [13] studied the systematic evaluation of the use of AI in medical education, and found that AI was used for learning support, student assessment, and course review; AI also provides students with instructional learning approaches and personalized feedback [14] or uses a neural network approach to analyze students' academic performance, thus, enabling teachers to make more representative assessments of students' knowledge according to the courses [15]. Let the machine not only carry out instructions, but also further learn human thinking, and this kind of wisdom development has never stopped in the past 50 years. Limited by the resources in the past, AI was difficult to apply in practice; however, with the great leaps forward of information technology in recent years, the maturity of AI technology can finally be implemented, and thus, has become a hot topic of discussion regarding hardware, computing efficiency, and algorithm refinement. Many enterprises regard AI as the next opportunity to create competitive advantages, and in the field of education, it has gradually become the trend of computer teaching development, promoting the reform of education and the process of teaching. Ref [16] pointed out that the combination of AI and education significantly improved teaching quality and developed new teaching strategies. Not only do teachers benefit from "smart systems" that help assess, collect data, improve learning progress, and develop new teaching strategies, students also benefit from "smart mentors" and "asynchronous learning" to improve learning outcomes. The combination of AI and education is conducive to the transformation of education, and provides the breakthrough of human knowledge, perception, and culture. Therefore, the application of AI in the field of education has become the focus of information technology and education research.

### 2.3. Application of Python in Education

Since Python is an object-oriented, interpreted programming language, and provides cross-platform versions suitable for Linux, Mac, and Windows [17], it has become a popular universal coding language in recent years. Moreover, it is an entry-level programming

language for computer science courses, as well as one of the most popular programming languages in both industry and academia [18].

Although Python is an interpretative language and its computing speed is not as fast as C, Python can flexibly use C or C++ to write extension suite modules. Its advantages are free open source, strong community support, readability of code, and simple syntax, thus, even beginners can learn easily. Secondly, each function library has its own namespace (named modules in Python), which can be nested in a very modular manner, and has various programming styles, such as programmatic, functional, imperative, and object-oriented programming. In addition, Python has a highly interactive program design that gradually executes program codes, which is very convenient for learning program design, debugging, and experimentation. It is widely used in scientific computing, image processing, web crawler, natural language processing, WEB development, big data, data mining, artificial intelligence, robot learning, visual teaching, data analysis, etc. [19–23].

### 2.4. Application of Deep Learning in Education

Since the advent of AI technology, it has been booming, plays an important role in our daily life, and completely changes our thinking, behavior, and interaction mode, especially with the emergence of artificial neural networks (ANN) and deep learning (DL) [24]. Deep learning is not a new technology; as early as the 1960s and 1970s, inspired by the biological nervous system, information scientists put forward the concept of multilayer artificial neural networks, and hoped that by simulating the biological nervous system, computers could achieve high intelligence like human beings. However, due to the poor performance of the hardware and the lack of software computing ability and digital data, there was no breakthrough. With the increasing popularity of Internet of Things (IOT) technology, the application level of AI deep learning is also becoming more and more popular in our daily life. The principle of deep learning is to treat information as if it were processed by the human brain and enhance traditional neural networks through a series of hidden layers to improve their predictive ability [25]. Ref [26] pointed out that deep learning has complex structures and relationships in high-dimensional data, and can dynamically construct new task-specific attributes from data representation [27], which enables deep learning models to surpass existing machine learning methods [28]; for example, deep learning has been successfully applied to visual and image processing [29], speech recognition [30], traffic control, power [31] and energy consumption [32], business credit scoring [33], drug molecular analysis [34], building energy management [35], natural language processing [36], and medical image interpretation [37]. In addition, regarding education, the successful prediction of high-risk dropout rate students through deep learning technology will facilitate timely intervention and implementation of corrective strategies and provide support and guidance for students in learning analysis [38].

### 2.5. Technology Acceptance Model

When innovative technology products come out, they all hope to be accepted and adopted by the public. The most widely used theory about the use of new technology products or innovative services is the technology acceptance model (TAM), as proposed by Davis (1986). According to the systematic reviews of 42 studies on the acceptance of e-learning by [39], the "technology acceptance model (TAM)" is the most popular theory in e-learning research, and 86% of such studies take the "technology acceptance model" as the theoretical basis [40]. Many studies have demonstrated that "perceived usefulness" and "perceived ease of use" directly affect learners' willingness to use e-learning [41–45].

According to [46], students' attitudes and behavioral intentions to using interactive whiteboard technology were verified by the extended technology acceptance model with interactivity and self-efficacy. The results showed that students' attitudes and behavioral use intentions are affected by interactivity, perceived self-efficacy, perceived ease of use, and perceived usefulness. According to the research results of [47], the technology acceptance

model is most affected by five external factors: self-efficacy, subjective norms, enjoyment, computer anxiety, and previous experience.

This study combined traditional classroom face-to-face teaching with online e-learning to construct the teaching strategy under the 4D diamond design to carry out teaching of the AIOT implementation course for senior high school students. Its theoretical basis is the "technology acceptance model", and self-efficacy and learning anxiety are external factors to explore students' learning outcomes and learning satisfaction after learning the AIOT course.

According to the above literature, the hypotheses of this study are, as follows:

- H1: External factor self-efficacy has a positive and significant impact on perceived ease of use.
- H2: External factor learning anxiety has a positive and significant impact on perceived ease of use.
- H3: External factor self-efficacy has a positive and significant impact on perceived usefulness.
- H4: External factor learning anxiety has a positive and significant impact on perceived usefulness.
- H5: Perceived ease of use has a positive and significant impact on perceived usefulness.
- H6: Perceived ease of use has a positive and significant impact on learning engagement.
- H7: Perceived usefulness has a positive and significant impact on learning engagement.
- H8: Learning engagement has a positive and significant impact on behavioral intention.
- H9: Behavioral intention has a positive and significant impact on learning outcomes.
- H10: Behavioral intention has a positive and significant impact on learning satisfaction.

## 3. Research Method and Design

The subject of this study is a class of a first grade high school in Kaohsiung City that was intentionally sampled to conduct an AIOT practical course experiment. The questionnaire, curriculum and teaching development, model assumptions, definitions and measurement methods, and research objects and instruments are discussed here.

### 3.1. Questionnaire Dimensions

According to the results of the literature analysis, the questionnaire included the external factors "self-efficacy" and "learning anxiety", and the outcome factors "learning outcomes" and "learning satisfaction", and "learning engagement", which influenced the technology acceptance model, in addition to the technology acceptance model of Davis (1986).

### 3.1.1. Self-Efficacy

When one believes that one can accomplish tasks independently, one will face and solve difficulties with a voluntary, efficient, and positive attitude [48,49]. The concept of self-efficacy is derived from Bandura's social learning theory, which is based on interactive determinism. In terms of the interaction between individual behaviors and the environment, individual beliefs will affect individual behaviors and adaptation to the environment.

IOT or ICT (information and communication technology) self-efficacy refers to the individual's perceived confidence in their computer, network, and other related abilities and knowledge [50–52]. The previous literature found that self-efficacy plays an important role in programming learning; for example, [47] investigated the interaction between visual coding (VPL) and self-efficacy, and found that VPL learning has a significant impact on learners' self-efficacy. In addition to reducing the negative attitude towards the difficulties encountered in the learning process, VPL can more effectively improve learners' self-efficacy. In 41 recent studies, [47] found that self-efficacy (SE) had a significantly positive impact on perceived ease of use (PEOU). Ref [53] found that computer self-efficacy was significantly positively correlated with the perceived ease of use of digital portfolios.

### 3.1.2. Learning Anxiety

According to the theory and evidence of [54], computer anxiety (CA) is associated with avoidance or less use of e-learning [41,42,55,56]. Computer anxiety is defined as a general feeling of unease, anxiety, or fear regarding one's current or future use of computers [57]. Computer anxiety has a negative and significant impact on students' intention to use e-learning systems [58,59]. Similarly, teachers who are anxious about using computers may also be reluctant to use e-learning systems [55], thus, it is very important to adopt effective strategies to reduce students' anxiety and cognitive burden in the learning process [60–62]. Therefore, learning anxiety has also been included as one of the important factors in the study of e-learning.

### 3.1.3. Learning Engagement

In MOOCs, the most common indicator to measure learning outcomes is the degree of learning engagement, which is the process of learners' continuous efforts to achieve their learning goals [63]. MOOC students also participate in various levels of activities, most often in the behavioral aspect [64–66], or their learning engagement is measured by watching teaching videos, posting opinions in discussion boards, taking tests, and completing tasks [67].

While the learning process of MOOCs can effectively promote learning engagement, it is necessary to consider the emotion, perception, and behavior of learning engagement; learning engagement usually consists of multiple factors (motivation, perception, and emotion) [68]. Most MOOC studies divide learning engagement into behavior, emotion, and perception [68,69]. Behavioral engagement refers to learners' asking questions and participating in discussions in the MOOC learning process. Emotional engagement refers to the degree to which learners are positive towards teachers, peers, and MOOC learning. Perceived engagement refers to learners' efforts to acquire complex knowledge or skills in the MOOC learning process [70].

According to [71], learning engagement was also defined by some scholars as "the energy used by students in the learning community, which is observed by behavioral, perceptive, or emotional indicators, and is impacted by a series of structural and internal factors, including mutual relations, interaction between learning activities, and learning environment". The more students are engaged and empowered in the learning community, the more likely they are to channel learning energy back into their learning, thus, producing a series of short-term and long-term learning outcomes that promote learning engagement [72].

### 3.1.4. Learning Outcomes

Learning outcomes mainly depend on understanding the overall benefits of learning [73], and are evaluated or measured according to learning objectives and teaching methods [74]. The traditional teaching-centered approach is measured by scores, which are related to learning and are usually grades or levels. The situational teaching method is learner-centered, which holds that teaching is not knowledge transfer, but the knowledge construction of learners. Social constructivism holds that knowledge is constructed through interpersonal interactions [75], thus, the evaluation of collaborative learning should integrate individual evaluation and peer evaluation. The practical inquiry model combines the shared world and the private world to realize, construct, and confirm meaning through self-reflection and shared discussions [76]. Another teaching design function in flipped teaching is to add group works [77]. Ref [78] pointed out that group learning through learning activities (such as paired sharing, paired problem solving assignments, and group discussions) under flipped teaching can improve learning outcomes. In addition, [79] pointed out that group learning activities can improve the effectiveness of the flipped classroom (mathematics). To sum up, [80] suggested that learning outcomes should include peer evaluation, self-evaluation, formative evaluation, and summative evaluation.

### 3.1.5. Learning Satisfaction

According to [81], learning satisfaction is an indicator of whether learners are satisfied with their learning experience. Ref [82] pointed out that their research results on learning satisfaction showed that there were many factors affecting learning satisfaction, such as online and face-to-face classroom interactions, feedback, students' and teachers' participation behavior, teaching activities, teaching content, online discussion, teaching skills, students' learning style, self-efficacy, and demographic characteristics. The research of [83] also showed that the rapid development of digital technology makes the diversification of teaching methods (e-learning, serious games, virtual simulation, etc.) possible, and that digital education seems to significantly improve theoretical and practical learning satisfaction. Ref [84] believed that understanding students' learning satisfaction is important for effective course design and learning outcomes [85]. Ref [86] found in their meta-analysis of mixed learning that adding tests had a positive effect on learning outcomes and satisfaction with mixed learning. The research by [87] also reached the same conclusion, meaning that an online learning system will improve learning satisfaction if it can provide multiple ways to evaluate learning and increase learners' interactions with each other. Therefore, in this study, learning satisfaction refers to the degree to which learners are satisfied with the learning process of learning AIOT in flipped teaching.

To sum up, flipped teaching can improve students' engagement, motivation, and satisfaction in the classroom [88–90]. Ref [91] pointed out that, compared with the control group, the experimental group of the flipped class mode (FCM) was significantly improved in satisfaction, and created and cultivated students' sense of achievement by using FCM, improved their participation motivation in the learning process, and established an active learning environment. Their study also further analyzed the learning satisfaction of students from the three groups of low, middle, and high scores in the experimental group, and the results showed that the satisfaction of students from the low score group was significantly higher than that from the middle score group and high score group.

### 3.2. Course Content and Teaching Design

While AIOT is gradually entering our daily life, the information technology courses taught in senior high school are still limited to the introduction of concepts, and there is a lack of opportunities for students to implement and experience how AIOT can be implemented and applied to all walks of life. Therefore, this study conceived the course content and teaching design through the double diamond design model, and combined theoretical courses and implementation courses, in order to deepen the basic concepts of senior high school students in AIOT. The course plan is shown in Figure 1.

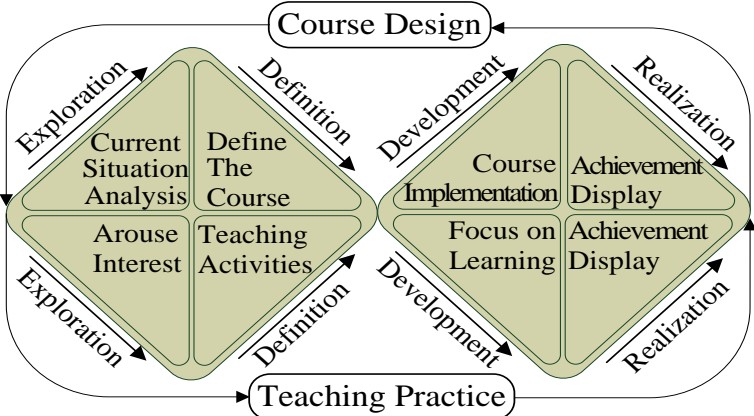

**Figure 1.** Course planning.

### 3.2.1. Course Design

In the curriculum design section, the course content was constructed according to the 4D double diamond design model process, and its stages were defined individually with this study as Stage 1. Exploration: current situation analysis; Stage 2. Definition: define the course; Stage 3. Development: course implementation; Stage 4. Achievement display, as follows.

1. Stage 1. Exploration: Current situation analysis

The rapid development of information and communication technology, faster network speed, stronger computing power, and technological innovations at different levels drove the revolution of industry 4.0; however, the teaching scene has always been unable to keep pace with the times, resulting in the phenomenon of a gap between production and learning, especially in senior high school. In order to shorten the transition period of talent training in the industry, how should senior high schools plan courses and develop teaching models in the field of science and technology? The theme of the experimental teaching in this study focused on how course planning can connect with the promotion of emerging science and technology-related courses in colleges and universities, how to meet the needs of industry, and how such courses can become characteristic in schools to attract students to study. Therefore, this study explored the future implementation trend and course content according to the current development of industry, government, and university, in order to help students understand the basic concepts of AIOT, and no longer regard AIOT as the plot of a science fiction film, but as a technology applied in industry, and then, integrate it into their daily life, as shown in Figure 2.

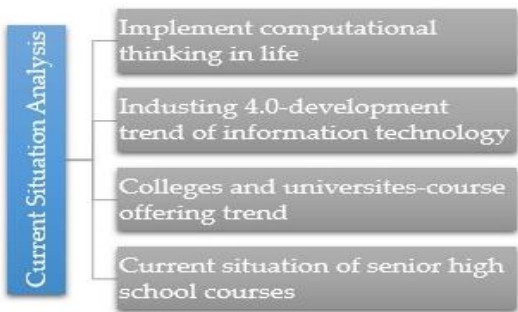

**Figure 2.** Current situation analysis.

2. Stage 2. Definition: Define the course

According to the current situation analysis in the exploration stage, it is confirmed that the course content of this experimental teaching can meet the current situation of the three aspects of the previous stage. First, according to the current trend of the application of emerging technologies in industry and the level of talent demand, course content is considered to be the basic knowledge and ability that senior high school students must cultivate. Secondly, it checked the course information network of colleges and universities and found the numbers of programming language courses (16, 55, 95) and schools (11, 18, 30) in emerging technologies in the past three years have been increasing year by year. Finally, in order to become a school-based characteristic course, the course design should be able to meet the requirements of implementing science and technology literacy, as well as the computational thinking of the 108-course outline, as shown in Figure 3.

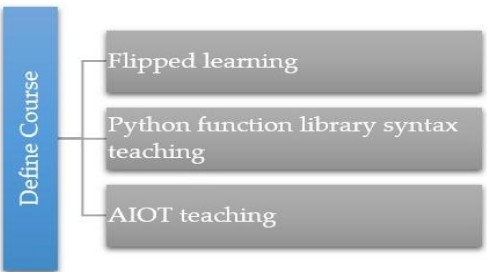

**Figure 3.** Define the course. AIOT: artificial intelligence of things.

To sum up, experimental teaching considered the AIOT development trend of industry 4.0 and planned the course outline. The course content is in line with the programming language widely used by industries, colleges, and universities in the development of AIOT to conduct Python language teaching, which will enable students to understand basic grammar and coding abilities. Finally, in order to avoid the traditional teaching method with poor learning outcomes, the SPOC teaching method is adopted in teaching skills, in order to arouse students' interest in active learning and further improve their learning outcomes.

3. Stage 3. Development: Course implementation

This study planned the course content after determining the course outline in the definition stage. In addition to traditional classroom presentations, existing online instructional videos that are relevant to the course content and no longer than 15 min are available. Secondly, it invited practitioners to join the teaching team for collaborative teaching, and relied on practitioners' practical experience to make students understand the real needs and pulse of the industry, which not only strengthened students' theoretical knowledge and practical application, but also gave play to the function of both industry experts and teachers, thus, enhancing the value of education, realizing the spirit of STEM by emphasizing "learning by doing, and doing by learning", and implementing the characteristics of "practical education". The teachers' group listed books related to AIOT, and students could choose a book according to their own interests, thus, enriching their scientific and technological literacy. In order to ensure the effective implementation of the courses in each unit, learning lists, learning experience sharing, and group reports were designed for each unit, which could facilitate understanding of the students' learning situation, and then, adjust the course content according to their situation. At the end of the experimental course, the SPOC–AIOT questionnaire was filled in to understand the students' learning outcomes and course satisfaction, as shown in Figure 4.

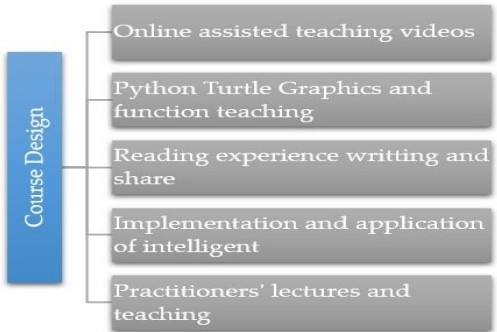

**Figure 4.** Course design.

4. Stage 4. Realization: Achievement display

In order to increase students' learning engagement, implementation courses were arranged to enhance students' learning interest. The implementation types were tailored to

the student's learning subject, thus, their professional knowledge could be combined with the experimental course to maximize the comprehensive effect. Therefore, in the course design with the intelligent speaker teaching suite as the medium, the practitioners led the students to gradually complete the setting process of raspberry AI, the intelligent speaker, and Chinese pronunciation, in order for the students to experience the conversational ability of AI natural language. Secondly, it combined the soldering practice of relay, DuPont line, and other parts with the pin position, and finally, completed the simulation of an intelligent speaker voice remote-controlled light switch, in order that students could understand the principle and practical application of AIOT, as shown in Figure 5.

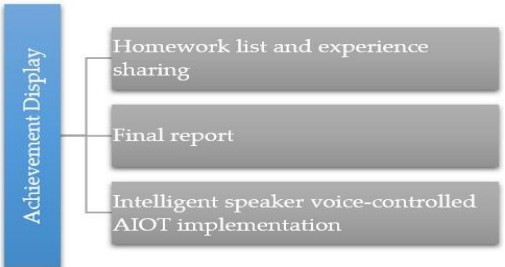

**Figure 5.** Achievement display.

3.2.2. Teaching Practice

In the teaching practice section, the course content was constructed according to the 4D double diamond design model process, and its stages were defined individually with this study as stage 1. Exploration: arouse interest; stage 2. Definition: teaching activities; stage 3. Development: focus on learning; stage 4. Realization: achievements display, as follows:

1. Stage 1. Exploration: Arouse Interest

To enable students to have resonance and actively participate in the course, this stage was mainly to arouse learning motivation; therefore, this experiment adopted group cooperative learning. In order to keep the course interesting for students, Python grammar and coding were not directly introduced during teaching, instead, the teaching activity of the "Turtle Graphics" course was adopted to make students understand the basic objects and functional instructions of the Python language through the process of drawing geometric figures. With the help of the concept of virtual codes, the intuitive compiling and drawing steps were conducted to help students quickly and thoroughly understand the concept of re-importing the programming language code, and students could repeatedly practice the original Turtle Graphics homework of each group with cyclic grammar, thus, the students would not become frustrated and give up learning immediately. In order to save time in class, online videos were provided for students to browse as a teaching aid.

The teaching process of this experiment attached importance to the construction of the students' learning process. In order to train students to record their learning process, and thus, enrich their learning process, the teachers' group listed books related to AIOT for students to borrow, and encouraged them write about their experience. Then, experienced judges of the national reading experience competition for middle school students reviewed their reading experience and gave suggestions. After the students revised their reading experience according to the suggestions, they participated in the reading experience writing competition for middle school students again.

2. Stage 2. Definition: Teaching Activities

The teaching activities in this stage focused on teaching basic functional grammar coding, thus, if traditional classroom teaching was used, it would not attract students' attention to take the initiative to learn. Therefore, the teaching focus was on the students' learning process, which was supplemented by related online videos and sharing activities between groups, rather than individual learning. In this stage of the learning process,

students were both learners and teachers, and their learning outcomes were improved through this dual role learning mode. In the course of functional grammar, in addition to learning the basic functions, the explanations of the functions that could be used in the intelligent speaker program code were also strengthened. At that time, the practitioners led the students to carry out intelligent speaker-related courses, in order that they could understand the principles and applications.

3.  Stage 3. Development: Focus on Learning

In order to break away from the traditional teaching method, this stage continued the mixed teaching and shared learning of the previous stage, and introduced experienced practitioners to lead students to practice, thus, turning theory into practice and strengthening students' concepts. The teaching object of this experiment was the students of an electrical and electronics group in a technology senior high school. In order to apply the programming language learned in this course during implementation, two major topics were planned in the implementation section under the discussion of teachers and practitioners. The first topic was the setting operation of the intelligent speaker, which helped students feel the experience of AI natural language conversation. The second topic was to continue the teaching activity of the first topic, take the intelligent speaker as the carrier, solder pins on the main board of the raspberry PI, and connect the relay to the correct pin position on the main board with a DuPont line. As this process was limited by the time that the practitioner taught at the school, a video of the practitioner instructing a group of students on how to solder and connect was shot in advance. By sharing through the class communication group, students of other groups could finish this part outside the classroom, thus, saving classroom time to focus on the teaching subject. This professional operation skill was the specialty area of the experimental subjects, thus, the implementation process could be felt more, and give full play to group cooperation and the participation of each member. Finally, practitioners introduced the principle and application of the grammar, in order that students could simulate the process of turning on and off the lights of AIOT intelligent home appliances through the natural language of the intelligent speaker.

4.  Stage 4. Realization: Achievement Display

In this stage, various achievements were displayed for each group, including their individual reading experiences and the geometric figure drawings of the groups (comparison of drawing instructions and circular instructions), which were presented according to creativity, program optimization, design concept, and visual aesthetics. In the implementation stage, practitioners and the teachers' group were invited to score according to the scoring items, such as function and team cooperation.

The timeline of the teaching courses developed in the four stages of the teaching practice is summarized in Figure 6.

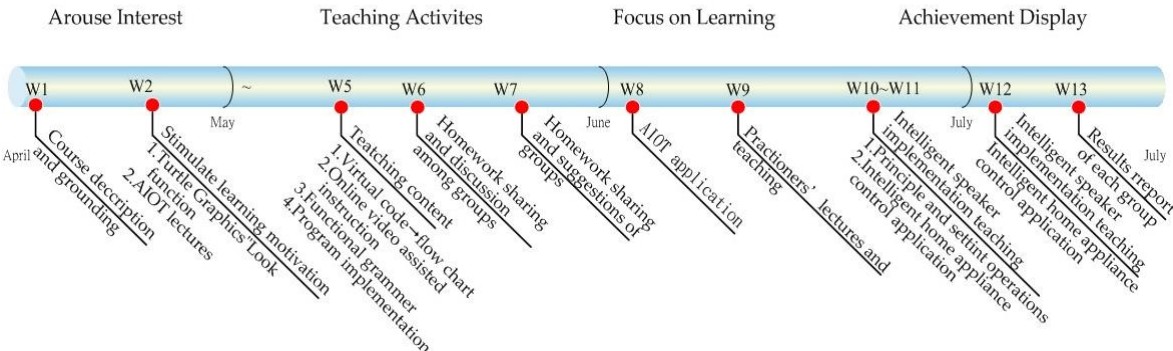

**Figure 6.** Timeline of teaching courses.

### 3.2.3. Research Model Hypothesis

According to the above research hypotheses, this study adopted the quasi-experimental research method and applied the small private online course (SPOC) to implement flipped teaching in a senior high school information technology AI and Internet of Things (AIOT) course to explore the learning outcomes and learning satisfaction with the experimental course. The SPOC–AIOT research framework is shown in Figure 7.

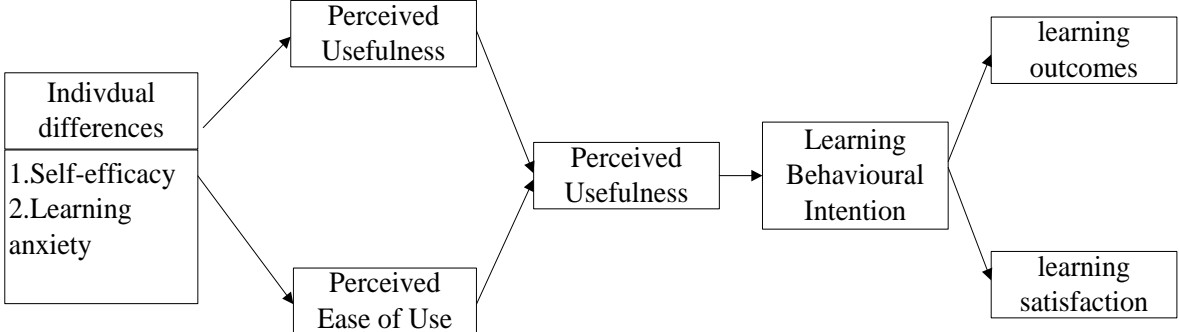

**Figure 7.** SPOC–AIOT research framework.

### 3.2.4. Operation-Type Definitions and Measurement Method

In this study, "the application of flipped teaching in a senior high school AIOT characteristic course" was explored using the questionnaire survey method for quantitative research with qualitative research verification. The scale was measured by a five-point Likert scale, where 1 point indicates strongly disagree, while 5 points indicates strongly agree. According to the above literature and research, the operation-type definitions for each dimension are summarized in Table 1.

**Table 1.** Operation-type definitions for each dimension of the SPOC–AIOT learning scale.

| Variable Name | Operation-Type Definition |
|---|---|
| Self-efficacy (se) | Learners' perceptive confidence in learning AIOT in computer, network, and other related abilities and knowledge |
| Learning Anxiety (la) | Learners generally feel uneasy, anxious, or afraid of using a computer to learn AIOT at present or in the future |
| Perceived Ease of Use (peu) | The degree of ease of use for learners to learn and perceive science and technology in AIOT |
| Perceived Usefulness (pe) | The degree to which learners believe that using technology to learn AIOT will improve their performance or save effort |
| Learning Engagement (le) | The process by which learners make continuous efforts to achieve their goal of learning AIOT |
| Behavioral Intention (bi) | The intensity of learners' willingness to use information systems to learn AIOT |
| Learning Outcomes (lo) | The knowledge and abilities acquired by learners after AIOT course or degree |
| Learning Satisfaction (ls) | Satisfaction and happiness obtained by learners in all aspects of teaching services when learning AIOT |

### 3.2.5. Research Subjects

A total of 36 Senior One students in Electrical Engineering from a technology senior high school in Kaohsiung were selected as the research subjects. The duration of the experiment was 13 weeks with two classes per week. In addition to collecting students' interactive learning and homework lists, after the course experiment, students filled in the "SPOC–AIOT learning scale" questionnaire, in order to observe, record, and collect the correlation between students' self-efficacy, learning anxiety, learning outcomes, and learning satisfaction with the AIOT course through the technology acceptance model during the flipped learning process.

### 3.2.6. Research Tools

Due to its experimental nature, this study was limited by the small sample size. To effectively verify its verification capability, the SmartPLS statistical software developed by [92] was used for analysis. This statistical software is not limited by variable assignment type or sample size, and has the advantages of good prediction and ability. In this study, 5000 samples were sampled repeatedly by the bootstrap resampling method as the parameter estimation and inference to solve the problem of small sample size, and to verify the relationship between various influencing factors, learning outcomes, and learning satisfaction.

## 4. Research Results

After the teaching experiment, the results of this study were analyzed in a quantitative and qualitative manner. Therefore, the background of the sample was analyzed first, then the methodology of this study was explained, and finally the data were analyzed by SmartPLS statistical software.

### 4.1. Sample Background Analysis

The sample of this study is Senior One students in Electrical Engineering from a technology senior high school in Kaohsiung. Based on the analysis of the basic data of the questionnaire, it can be seen that most of the students had never accessed AIOT-related courses or Python programming language before this experimental course. Under the experimental teaching method, the statistics are shown in Table 2.

**Table 2.** Sample descriptive statistics.

| Items | Number | Percentage |
|---|---|---|
| Have you ever taught yourself any courses or procedures related to AI deep learning before this course? | No: 32 | 89% |
| | Yes: 4 | 11% |
| Times of searching for AI deep learning-related courses on the internet every week | 0: 23 | 64% |
| | 1–3 times: 13 | 36% |
| Average time of daily online browsing of AI deep learning-related courses | Within 30 | 100% |
| | minutes: 36 | 0% |
| Number of group discussions per week | 0: 5 | 14% |
| | 1~3 times: 28 | 78% |
| | 4~6 times: 3 | 8% |

As can be seen from Table 3, before the course experiment, students had little foundation or understanding of AI and IOT, but after the experimental flipped teaching by SPOC, the proportion of students learning through discussion and sharing increased significantly.

### 4.2. Research Method

The samples in this study were divided into two orders: the measurement model analyzed the reliability and validity of the model, including "reliability of individual items", "internal consistency", "convergent validity", and "discriminant validity"; and "structural model analysis" was conducted to verify and estimate the "model explanatory power (R-square, R2)" and "path coefficient" of the structural model, and explore the causality between the dimensions.

The measurement model meets the following requirements: (1) the factor loading of each item must be greater than 0.5; (2) composite reliability and Cronbach's $\alpha$ should be at least greater than 0.5, and preferably greater than 0.7, to ensure the internal consistency of all dimensions; (3) the average variance extracted (AVE) must be greater than 0.5, and the square root of the AVE of each dimension must be greater than the correlation coefficient

of other dimensions; (4) this factor loading must be greater than other factor loadings, i.e., own-loadings are greater than cross-loadings, which means that the measurement has good convergent validity and discriminant validity.

The structural model is judged by the indicators of (1) whether the standardized path coefficient reaches statistical significance; (2) determines the model explanatory power by explanatory variance ($R^2$) and evaluated explanation effect value ($f^2$). Generally, when the $R^2$ value is close to 0.25, it can be regarded as slightly weak explanatory power; when it is close to 0.50, the model has moderate explanatory power; when it is close to 0.75, the model explanatory power is very significant. While the $f^2$ value is used to evaluate whether the exogenous variable has the explanatory power to the endogenous variable, the principle is that $0.02 < f^2 \leq 0.15$ is considered a small effect; $0.15 < f^2 \leq 0.35$ is considered a medium effect; and $f^2 > 0.35$ is considered a large effect [93].

According to the above statistical test process, this study proposes a conceptual model, as shown in Figure 8.

**Table 3.** Factor loadings of all dimensions of the SPOC–AIOT scale.

| Dimension | Item | Factor Loading (Out Loading) | Cronbach's $\alpha$ | CR Value | AVE Value |
|---|---|---|---|---|---|
| Learning Anxiety | la_1 | 0.914 | 0.782 | 0.901 | 0.820 |
| | la_4 | 0.897 | | | |
| Behavioral intention | bi_2 | 0.909 | 0.766 | 0.895 | 0.810 |
| | bi_5 | 0.891 | | | |
| Learning outcomes | lo_1 | 0.912 | 0.756 | 0.861 | 0.675 |
| | lo_3 | 0.792 | | | |
| | lo_6 | 0.751 | | | |
| Learning engagement | le_1 | 0.847 | 0.860 | 0.905 | 0.704 |
| | le_2 | 0.776 | | | |
| | le_4 | 0.884 | | | |
| | le_6 | 0.846 | | | |
| Learning satisfaction | ls_2 | 0.797 | 0.815 | 0.891 | 0.732 |
| | ls_3 | 0.930 | | | |
| | ls_6 | 0.833 | | | |
| Perceived ease of use | peu_3 | 0.865 | 0.723 | 0.845 | 0.647 |
| | peu_4 | 0.832 | | | |
| | peu_5 | 0.707 | | | |
| Perceived usefulness | pu_1 | 0.833 | 0.750 | 0.855 | 0.664 |
| | pu_4 | 0.766 | | | |
| | pu_5 | 0.843 | | | |
| Self-efficacy | se_1 | 0.893 | 0.883 | 0.911 | 0.632 |
| | se_2 | 0.745 | | | |
| | se_3 | 0.855 | | | |
| | se_4 | 0.788 | | | |
| | se_5 | 0.725 | | | |
| | se_6 | 0.747 | | | |

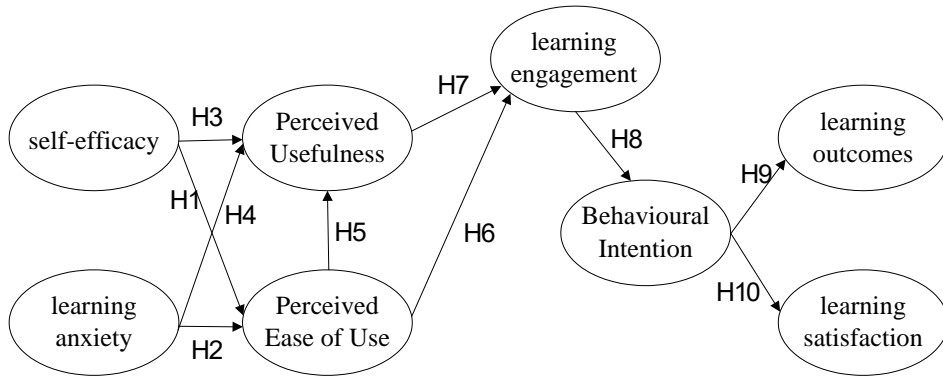

**Figure 8.** Conceptual model.

### 4.3. Data Analysis

#### 4.3.1. Verification of Common Method Variation

The questionnaires in this study are self-presentation scales. In order to avoid measurement error variation caused by homologous deviation, the suggestions of [94] were taken into consideration in the design and arrangement of the questionnaires, and precautionary measures were taken, such as the information concealment of interviewees, the meaning concealment of questions, the random allocation of questions, and the word organization of questions, in order to minimize the errors caused by common method variation.

#### 4.3.2. Measurement Model Analysis

1. Collinearity Evaluation Analysis

   In order to understand whether there is a collinearity problem among all variables, the variance inflation factor (VIF) was used for collinearity evaluation. According to the research of [95], when the VIF value is greater than 5, there will be a collinearity problem. After analysis, the items of la_2, bi_1, bi_3, lo_2, lo_5, le_3, ls_1, ls_5, peu_1, peu_2, and pu_2 with VIF values greater than 5 were deleted, while the VIF values of the remaining items were between 1.152 and 3.805, meaning all of them were within the range recommended by Hair et al. Therefore, there is no collinearity problem in this research model.

2. Reliability Analysis:
   - General reliability indicator: in order to verify the reliability of each item, reliability analysis (Cronbach's α) was performed on each dimension although the Cronbach's α value. The α values of the remaining dimensions ranged from 0.750 to 0.883, meaning all of them were above the minimum values of all thresholds, and thus, within the reliability range recommended by Nunnally, as shown in Table 3.
   - Composite reliability: in order to verify the consistency of the observed variables of each dimension, composite reliability testing was carried out, and the results show that the CR values of each dimension ranged from 0.845 to 0.911, which are all higher than the threshold value of 0.7 [96], as shown in Table 3.
   - Indicator reliability: considering factor loadings with the threshold value of 0.7 [95] as the basis for deleting items, the items of la_3, la_6, bi_4, ls_4, lo_4, le_5, and pu_3 were deleted in order, and the factor loadings of the other items were all between 0.707 and 0.930. In summary, it is shown that this questionnaire has good reliability, as shown in Table 3.

3. Construction Validity Analysis
   - Convergent validity: the purpose of convergent validity is to determine the consistency of each dimension, and the average variance extracted (AVE) value of each dimension was between 0.632 and 0.820. As shown in Table 3, the average

explanatory power of each dimension to the indicator is more than 50%, with good convergent validity.

- Discriminant validity: before structural model analysis, one of the prerequisites is to test discriminant validity among different dimensions, and the most commonly used items to test discriminant validity are cross-loading and the Formell–Larcker criterion. It can be seen from Table 4 that the factor loadings of each dimension are all greater than the cross-loading between the dimension and other dimensions. Secondly, according to the Formell–Larcker criterion, if the AVE square root of each dimension is greater than the correlation coefficient between the dimension and other dimensions, then discriminant validity can be achieved. The results show that the AVE square roots of each dimension are between 0.795 and 0.900, which are greater than the correlation coefficient between the dimension and other dimensions. As shown in Table 5, each dimension has good convergent validity, thus, it can be concluded that all the dimensions in this study have good construction validity.

**Table 4.** Cross-loading scale of all dimensions of the SPOC–AIOT scale.

| Dimension | Item | 1 [a] | 2 [b] | 3 [c] | 4 [d] | 5 [e] | 6 [f] | 7 [g] | 8 [h] |
|---|---|---|---|---|---|---|---|---|---|
| 1 | la_1 | 0.914 | | | | | | | |
| | la_4 | 0.897 | | | | | | | |
| 2 | bi_2 | | 0.909 | | | | | | |
| | bi_5 | | 0.891 | | | | | | |
| 3 | lo_1 | | | 0.912 | | | | | |
| | lo_3 | | | 0.792 | | | | | |
| | lo_6 | | | 0.751 | | | | | |
| 4 | le_1 | | | | 0.847 | | | | |
| | le_2 | | | | 0.776 | | | | |
| | le_4 | | | | 0.884 | | | | |
| | le_6 | | | | 0.846 | | | | |
| 5 | ls_2 | | | | | 0.797 | | | |
| | ls_3 | | | | | 0.930 | | | |
| | ls_6 | | | | | 0.833 | | | |
| 6 | peu_3 | | | | | | 0.865 | | |
| | peu_4 | | | | | | 0.832 | | |
| | peu_5 | | | | | | 0.707 | | |
| 7 | pu_1 | | | | | | | 0.833 | |
| | pu_4 | | | | | | | 0.766 | |
| | pu_5 | | | | | | | 0.843 | |
| 8 | se_1 | | | | | | | | 0.893 |
| | se_2 | | | | | | | | 0.745 |
| | se_3 | | | | | | | | 0.855 |
| | se_4 | | | | | | | | 0.788 |
| | se_5 | | | | | | | | 0.725 |
| | se_6 | | | | | | | | 0.747 |

a: learning anxiety, b: behavioral intention, c: learning outcomes, d: learning engagement, e: learning satisfaction, f: perceived ease of use, g: perceived usefulness, h: self-efficacy.

**Table 5.** Reliability analysis of all dimensions of the SPOC–AIOT scale.

| Dimension | | Formell–Larcker | | | | | | | |
|---|---|---|---|---|---|---|---|---|---|
| | | 1 | 2 | 3 | 4 | 5 | 6 | 7 | 8 |
| 1 | Behavioral intention | 0.900 | | | | | | | |
| 2 | Perceived usefulness | 0.738 | 0.815 | | | | | | |
| 3 | Perceived ease of use | 0.479 | 0.632 | 0.804 | | | | | |
| 4 | Learning anxiety | −0.268 | −0.147 | −0.189 | 0.906 | | | | |
| 5 | Learning engagement | 0.744 | 0.672 | 0.605 | −0.356 | 0.839 | | | |
| 6 | Learning outcomes | 0.488 | 0.493 | 0.310 | −0.360 | 0.321 | 0.821 | | |
| 7 | Learning satisfaction | 0.718 | 0.653 | 0.439 | −0.391 | 0.684 | 0.630 | 0.856 | |
| 8 | Self-efficacy | 0.589 | 0.686 | 0.597 | −0.157 | 0.593 | 0.423 | 0.626 | 0.795 |

To sum up the analysis results, the measurement model of this study was tested by collinearity analysis, reliability analysis, and construction validity, and the results all meet the academic requirements, representing that the measurement systems of the eight dimensions in the SPOC–ATIO scale have reliability, convergent validity, and discriminant validity. Therefore, structural model analysis can be performed to test the causal path relationship among the dimensions.

### 4.3.3. Structural Model Analysis

This study used SmartPLS3 to analyze the path analysis among the various dimensions of the research framework, and bootstrapping was used to conduct path analysis and test the research hypotheses through the resampling method for 5000 times. The research framework is a single definite directional relation; therefore, two-tailed testing was used in this study, and the significance level (*p* value) less than 0.05 was taken as the judgment standard [97,98]. The results of the statistical verification of the overall model are shown in Figure 9.

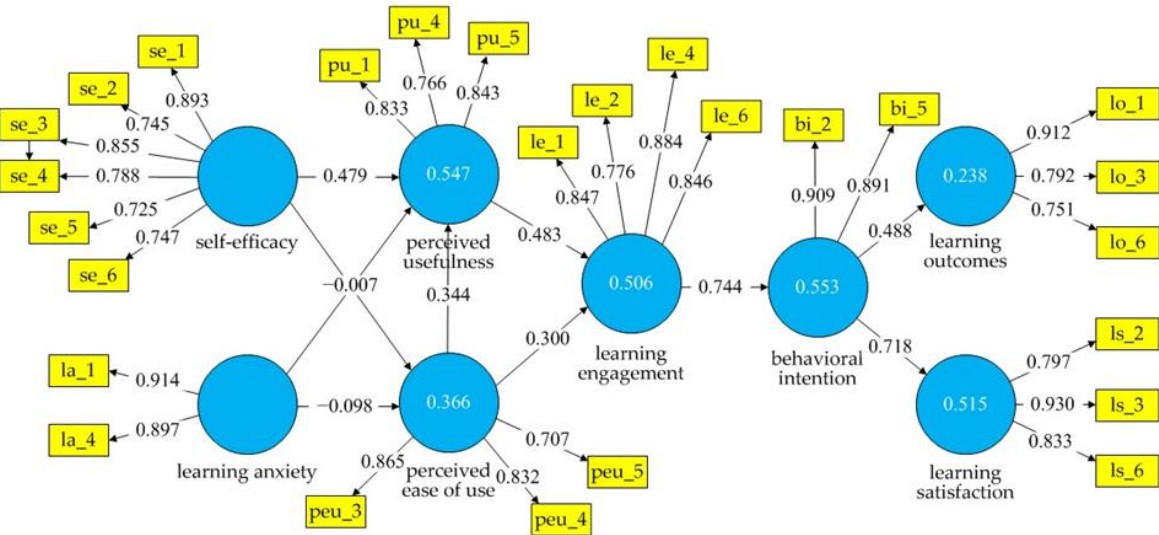

**Figure 9.** Overall model of the SPOC–AIOT scale.

According to [95], a systematic method was proposed for the evaluation of the structural model. The process was divided into the stages of collinearity diagnosis of the structural model, significance testing of the path coefficient, evaluation of $R^2$, and evaluation of explanation effect value $f^2$.

Wherein, the VIF values of all indicators of the structural model are less than 5. According to the inner VIF values, the VIF values between the related dimensions are also less than the threshold value of 5, indicating that the collinearity problem between the indi-

cators and dimensions in the measurement model and structural model is not serious [95], and the collinearity problem will not adversely affect the path coefficient estimation of the structural model in the future. The test results are shown in Tables 6 and 7.

Secondly, the significance test of the path coefficient was to evaluate the size of $R^2$ and evaluate the explanation effect value $f^2$, and the results are summarized in Table 7.

1.  Verification of path relationship

The path coefficients of the hypothetical relationships of H1 (Self-efficacy → Perceived ease of use), H3 (Self-efficacy → Perceived usefulness), H5 (Perceived ease of use → Perceived usefulness), H6 (Perceived ease of use → Learning engagement), H7 (Perceived usefulness → Learning engagement), H8 (Learning engagement → Behavioral intention), H9 (Behavioral intention → Learning outcomes) and H10 (Behavioral intention → Learning satisfaction) in this research model are 0.582, 0.479, 0.344, 0.300, 0.483, 0.744, 0.488 and 0.718, respectively, which reach significance level and are tenable.

- Impact of self-efficacy on learning outcomes and learning satisfaction in implementing the flipped teaching of the AIOT course, as based on the technology acceptance model

**Table 6.** SPOC-AIOT scale structure model VIF verification table.

| | Dimension | Item | Variance Inflation Factor (VIF) | Inner VIF Values | | | | | | | |
|---|---|---|---|---|---|---|---|---|---|---|---|
| | | | | 1 | 2 | 3 | 4 | 5 | 6 | 7 | 8 |
| 1 | Learning anxiety | la_1<br>la_4 | 1.699<br>1.699 | | 1.000 | | | | 1.025 | | |
| 2 | Behavioral intention | bi_2<br>bi_5 | 1.625<br>1.625 | | | | 1.000 | | | | |
| 3 | Learning outcomes | lo_1<br>lo_3<br>lo_6 | 2.125<br>1.577<br>1.521 | | 1.000 | | | | | | |
| 4 | Learning engagement | le_1<br>le_2<br>le_4<br>le_6 | 2.572<br>1.714<br>2.288<br>2.481 | | | | | | | | |
| 5 | Learning satisfaction | ls_2<br>ls_3<br>ls_6 | 1.786<br>2.853<br>1.957 | | | | | | | | |
| 6 | Perceived ease of use | peu_3<br>peu_4<br>peu_5 | 2.532<br>2.372<br>1.152 | | | | 1.664 | | | 1.577 | |
| 7 | Perceived usefulness | pu_1<br>pu_4<br>pu_5 | 1.453<br>1.478<br>1.602 | 1.041 | | | 1.664 | | | | |
| 8 | Self-efficacy | se_1<br>se_2<br>se_3<br>se_4<br>se_5<br>se_6 | 3.805<br>1.789<br>2.920<br>3.119<br>2.408<br>2.102 | | | | | | 1.025 | 1.559 | |

**Table 7.** T structural model evaluation and verification table of the SPOC–AIOT scale.

| Hypothesis | Relationship | Path Coefficient | t Value | Decision | $R^2$ | $f^2$ | 95%CI LL | 95%CI UL | Fitness |
|---|---|---|---|---|---|---|---|---|---|
| H1 | Self-efficacy → Perceived ease of use | 0.582 * | 4.844 | True | 0.366 | 0.521 | 0.387 | 0.779 | |
| H2 | Learning anxiety → Perceived ease of use | −0.098 | 0.588 | False | | 0.015 | −0.375 | 0.170 | SRMR [a] = 0.122 NFI [b] = 0.409 RMS_theta [c] = 0.229 |
| H3 | Self-efficacy → Perceived usefulness | 0.479 * | 2.794 | True | 0.547 | 0.326 | 0.195 | 0.760 | |
| H4 | Learning anxiety →Perceived usefulness | −0.007 | 0.044 | False | | 0.000 | −0.303 | 0.234 | |
| H5 | Perceived ease of use → Perceived usefulness | 0.344 * | 2.284 | True | | 0.166 | 0.075 | 0.567 | |
| H6 | Perceived ease of use → Learning engagement | 0.300 * | 2.002 | True | 0.506 | 0.109 | 0.084 | 0.570 | |
| H7 | Perceived usefulness → Learning engagement | 0.483 * | 3.218 | True | | 0.284 | 0.214 | 0.704 | |
| H8 | Learning engagement → Behavioral intention | 0.744 * | 11.125 | True | 0.553 | 1.239 | 0.626 | 0.845 | |
| H9 | Behavioral intention → Learning outcomes | 0.488 * | 5.053 | True | 0.283 | 0.313 | 0.345 | 0.657 | |
| H10 | Behavioral intention → Learning satisfaction | 0.718 * | 8.337 | True | 0.518 | 1.063 | 0.553 | 0.837 | |

* indicates being significant at the significance level of 0.05. a: Standardized Root Mean Square Residual, b: Normed Fit Index, c: root mean squared residual covariance matrix of the outer model residuals

According to the quantitative research results, after the implementation of the AIOT characteristic course with flipped learning, self-efficacy can indeed improve students' learning outcomes. In other words, the higher the students' self-perceived confidence in computer use, the internet, and other information and communication-related abilities and knowledge, the more conducive to e-learning outcomes. Moreover, qualitative research shows that:

Students said that the use of online videos for self-learning is not limited by time and space, which facilitates repeated learning and further study. Compared with books, online videos can focus on the characteristics of the videos.

*As a computer user, I can concentrate more on computer videos than books.*

(20200107)

*I can make good use of my free time and let the videos match my timetable. Unlike before, I can keep learning through online videos, not limited by time and space, and I can also watch them over and over again, and review ideas when I don't understand them.*

(20200124)

*I was able to use online videos to understand what I didn't understand in class or to study further.*

(20200136)

The flipped learning mode of mixing traditional classroom teaching and online learning was used for teaching the AIOT course, and the results show that it can indeed improve students' learning satisfaction. In other words, the higher the students' perception of learning activities through various learning channels, the more helpful it is to improve students' learning satisfaction. Qualitative research shows that:

Students expressed their preference for online learning over traditional teaching, as online learning is convenient, interesting, not boring, and makes learning easier. The most important thing is that teachers cooperate with students, rather than students cooperate with teachers.

*I prefer online learning to traditional teaching, because the traditional teaching method is not easy to understand and boring. In addition to being easier to understand, online learning is less boring than traditional teaching.*

(20200309)

*I prefer online learning to traditional teaching, because modern online learning is more convenient and interesting, and the traditional teaching method seems a little old-fashioned. I hope I can use online learning more.*

(20200301)

*In traditional teaching, we cooperate with teachers to learn. But online teaching allows teachers to work with us, and when we don't understand something, we use online teaching videos to remove the confusion again and again, making learning easier.*

(20200324)

- Impact of learning anxiety on learning outcomes and learning satisfaction in implementing flipped teaching of the AIOT course, as based on the technology acceptance model

According to the quantitative research results of the technology acceptance model, learning anxiety has no significant impact on perceived usefulness or perceived ease of use. In other words, students do not have anxiety about learning new knowledge with digital technology, which may be due to the experimental teaching process, as it reduces students' learning anxiety by traditional classroom teaching and group experience sharing, although learning is carried out by flipped teaching. Therefore, students' learning outcomes and learning satisfaction under the flipped learning mode are strengthened through the technology acceptance model. Qualitative research shows that:

Students said that the reason they do not feel anxious about learning new knowledge using digital technology is because online videos can be watched repeatedly. Young people prefer 3C products, and it is easy to get started, as they are not limited by space and can learn online at any time.

*It helps more or less. If you don't understand the online videos, you can go back to watching them again; if you still don't understand it, you can directly check it online. If you still don't understand it, you can also ask the teacher or expert in the professional field online.*

(20200208)

*Modern teenagers prefer 3C products. There are many advantages, such as check-in, uploading videos, sharing websites, etc., but the disadvantage is that we can't know whether we are interacting with real people online.*

(20200213)

*I prefer online learning. Because when you don't understand a chapter or a segment of audio and video teaching, you can watch it repeatedly, and you are not restricted by the fixed location of the class.*

(20200230)

- Impact of learning engagement on learning outcomes and learning satisfaction in implementing flipped teaching of the AIOT course, as based on the technology acceptance model

According to the quantitative research results, learning engagement is significantly affected by perceived usefulness and perceived ease of use in the technology acceptance model, and will affect students' attitude and perception towards the use of information and communication technology. In other words, in the process of learning new knowledge, students believe that digital technology is positively conducive to self-learning and easy to use, and will not cause learning burden. They can actively participate in the course

learning situation and improve their willingness to use information technology, and thus, improve their learning outcome and satisfaction. Qualitative research shows that:

Students said that the reasons for digital technology affecting learning engagement include practical operation, practice, and experience, which makes it easier to understand the theory, application, and convenience of AI, which not only arouses interest in learning, it also stimulates the inspiration and the creation of AI.

> *It is no longer a rigid course, but a practical operation and practice, so that students can better understand the principles of AI and become interested in it at the same time.*

(20200411)

> *It allows us to personally experience the convenience of AI and helps us learn about it.*

(20200430)

> *For modern AI engineers and programmers, the actual operation and application can inspire more inspiration. As the saying goes, practice makes perfect. Only by continuous operation, repeated observation, and discussion, can we apply this knowledge to masterpieces.*

(20200413)

2. Evaluation of Model Explanatory Power

Generally, when the $R^2$ value is close to 0.25, it can be regarded as having slightly weak explanatory power; when it is close to 0.50, the model has moderate explanatory power; when it is close to 0.75, the model explanatory power is very significant (Hair et al., 2014). In this research model, according to the criterion, learning outcomes (0.238) belong to weak explanatory power; while the other dimensions, namely, perceived ease of use (0.366) belong to moderate explanatory power, perceived usefulness (0.547), learning engagement (0.506), behavioral intention (0.553), and learning satisfaction (0.515) belong to highly effective explanatory power. According to the $f^2$ value, as proposed by Cohen (1988), it evaluates whether an exogenous variable has significant explanatory power to an endogenous variable, which follows the principle that $0.02 < f^2 \leq 0.15$ is considered a small effect; $0.15 < f^2 \leq 0.35$ is considered a medium effect; and $f^2 > 0.35$ is considered a large effect. The research results show that there is highly effective explanatory power of self-efficacy towards perceived ease of use (0.521), learning engagement towards behavioral intention (1.239), and behavioral intention towards learning satisfaction (1.036); there is moderate explanatory power of perceived ease of use towards perceived usefulness (0.166), perceived usefulness towards learning engagement (0.284), behavioral intention towards learning outcomes (0.313) and self-efficacy towards perceived usefulness (0.326); there is weak explanatory power of learning anxiety towards perceived ease of use (0.015) and perceived ease of use towards learning engagement (0.109); and learning anxiety has no explanatory power towards perceived usefulness (0.000). On the whole, the explanatory power of the exogenous dimensions towards the endogenous dimensions is above the medium–high effect.

3. Evaluation of overall model fitness

The model test results in this study are SRMR = 0.122 > 0.08 and RMS_theta = 0.229 > 0.12. Although the overall model fitness is not good, in partial least squares structural equation modeling(PLS-SEM), the model fitness should be determined by the fitness results, as well as comprehensively evaluated by the significance of the path coefficient, model explanatory power, and prediction ability. Therefore, this model meets the requirements of academic overall model fitness.

## 5. Conclusions and Suggestions

### 5.1. Conclusions

This study conducted experimental teaching with an AIOT implemented teaching course, explored the 4D double diamond design course and the planning teaching content,

and observed the learning outcomes and satisfaction of students in implementing the AIOT course using the TAM proof of concept under individual differences, such as self-efficacy, anxiety, etc. The conclusions are, as follows:

### 5.1.1. The Impacts of Individual Differences on the Technology Acceptance Model

1.  The impact of self-efficacy on perceived ease of use and usefulness

In this experiment, self-efficacy was used as one of the external variables affecting the technology acceptance model, which refers to "the user's self-perception of computer ability for online learning", and the results show that self-efficacy plays an important role in the success of flipped learning, and "self-efficacy" has a significant positive effect on users' "perceived usefulness" and "perceived ease of use". Therefore, when the user's personal computer self-efficacy is better, and they agree that online learning is useful and easy to use, their willingness to use will be higher.

The empirical results of this study once again verify that the TAM theory, as proposed by [99], can specifically explain the relationship between users' "learning engagement", "perceived usefulness", "perceived ease of use", "willingness to use", and "self-efficacy" of using online e-learning in implementing flipped teaching.

2.  The impact of learning anxiety on perceived ease of use and usefulness

The results show that "self-anxiety" did not significantly predict the "perceived ease of use" or "perceived usefulness" of implementing flipped learning by using online e-learning, indicating that the students in this experiment did not have any anxiety about using digital technology to learn new knowledge about science and technology. The reason may be that the teaching process used mixed traditional classroom teaching and e-learning, thus, reducing the teaching mode of fully using online learning, and the learning experience of each group was shared in the course implementation process, which greatly reduced students' anxiety and unease about learning the programming language. The videos provided for students to browse were all available online teaching materials. The credibility of the content also reduced students' computer anxiety and provided them with a wealth of new knowledge about science and technology; therefore, instead of feeling anxious, students found it exciting to use computers and easy to use computer-aided learning. Thus, students were willing to use digital teaching materials to assist learning, which will increase their willingness and frequency of using digital teaching materials in the future.

### 5.1.2. Relationship between the Technology Acceptance Model and Learning Engagement

1.  The impacts of various dimensions of the technology acceptance model

The results show that "perceived ease of use" and "perceived usefulness" have a positive effect on the prediction of "behavioral intention" of flipped teaching using online digital teaching materials. In addition to verifying that current senior high school students have a considerable foundation to use information devices for learning, it also verifies that students feel that using digital teaching materials can improve their learning performance. This study also found that students attach importance to the ease of access and the browsing of online digital teaching materials, and whether the learning information provided is needed, correct, and useful. Therefore, when conducting experimental teaching, paying attention to online teaching materials and learning information will enhance students' willingness to continue to use digital teaching materials for independent learning. The experimental results also confirm that "perceived ease of use" has a positive and high degree of impact on "perceived usefulness".

2.  The impact of technology acceptance model and learning engagement

The results of the data analysis show that the higher the degree of students' "learning engagement", the more significant the positive correlation is found between their "behavioral intention" of using online learning for flipped teaching. The results of the

path analysis also show that students' "learning engagement" in flipped teaching has a significant impact on their "behavioral intention" towards online learning, thus, "learning engagement" positively affects students' actual "behavioral intention" towards learning. The more students experience the use of digital technology and information, the easier it is to use it and the more it helps them to improve themselves, and the more they will be involved in learning. Even if one has never accessed new knowledge about science and technology, such as AI and IOT, one can share them with peers providing one uses it through the simple connection of existing online teaching materials and the implementation of an experiential course, and teachers can lead students to experience the real application of AIOT in our everyday environment. These results suggest that AIOT will no longer be a distant technological fantasy.

3.  The impact of the technology acceptance model on learning outcomes and learning satisfaction

The results show that students are highly receptive to learning new knowledge about science and technology, and students' learning outcomes and satisfaction are deeply influenced by their attempts to learn through interactions with each other or through the implementation course of intelligent speaker applications, which can be verified by the path coefficient of structural model analysis. Wherein, the more willing students are to use digital technology to conduct AIOT-related courses, the higher their learning satisfaction will be. Secondly, the more students devote themselves to the learning environment of flipped teaching, the higher their intention of learning new knowledge about science and technology.

To sum up, the prototype of the SPOC–AIOT teaching framework, as proposed in this experimental teaching, is shown in Figure 10. The three regions are teachers, students, and e-learning as the main axis, and the intersection of the three regions is the flipped teaching of the AIOT course as the core. The intersection region of teachers and e-learning is self-made teaching materials or online teaching materials, as prepared by teachers using digital technology as supplementary teaching materials for students' traditional classroom learning. Secondly, the intersection region of students and e-learning is that students use e-learning for self-learning outside the classroom and share it with other students through digital equipment. The intersection region of teachers and students is that students and teachers carry out various teaching activities in the classroom. The arrow direction is intended to revise and adjust the course design and teaching practices according to the "4D diamond design" framework.

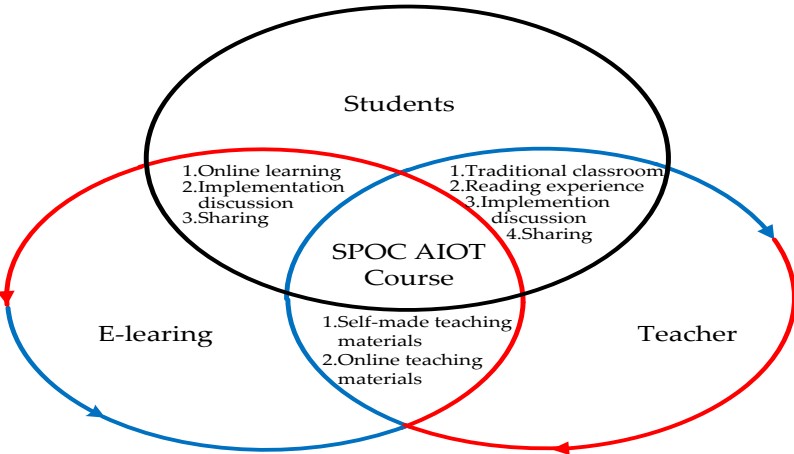

**Figure 10.** SPOC–AIOT teaching framework.

*5.2. Research Findings and Suggestions*

According to the research results, most students have never accessed AI and IOT-related knowledge before this experiment, thus, finding out how to integrate information

courses into related technology knowledge under the framework of the new course outline is a big challenge. Therefore, in terms of course conception, teaching material preparation, and teaching design, teachers should repeatedly explore and revise the content to provide students with easy-to-understand auxiliary learning materials and a simple learning platform to improve students' self-learning efficiency, in order that students can perceive the ease of use and usefulness of the software and hardware, and thus, reduce their learning anxiety when in contact with new knowledge or information literacy.

In addition to designing course content, teachers should guide students' learning engagement, thus, the focus of learning should be shifted from teachers to students. In terms of teaching, small-scale online learning courses are adopted for flipped teaching, and the teaching process requires students to share their experiences through their homework, and then, revise the homework content, in order to achieve the best learning outcomes and learning satisfaction.

In order to avoid teachers investing too much time in preparing digital teaching materials, it is suggested they search for relevant online teaching videos and provide students with links to browse. The videos should not be too long (within 5–15 min), in order that students will not lose interest in learning due to the time being too long. Finally, pure theory courses cannot attract students into a learning situation, thus, the best teaching mode is to combine the theoretical content of the course with physical objects and the implementation course, namely, to cooperate with the professional knowledge of technology senior high school students, which will surely improve the students' learning performance and learning quality. In the experimental teaching in this study, students with relevant background and skills in electrical engineering could experience deep learning language recognition, and simulate the opening and closing practices of intelligent home appliance switches through IOT with an intelligent speaker. In this experimental course, students showed their professional knowledge, such as how soldering, light-emitting diodes, relays, and other parts are connected to the intelligent speaker and bread board, which can actually realize cross-field learning.

*5.3. Research Limitations*

As this experimental study focused on the application of flipped teaching in the characteristic courses of senior high school AI deep learning, other characteristic courses are not explored. The Senior One students from a public senior high school in Kaohsiung were taken as the research subjects, thus, in order to infer the research results, various factors, such as region, students' socioeconomic background, school size, and grade, should be considered. This study conducted a quasi-experiment, and the research tools and reliability used in this study are only applicable to the experimental school, and are not applicable to other experimental methods, schools, or grades. In accordance with the actual teaching schedule, the experimental teaching lasted for 13 weeks, with two classes per week, for a total of 26 classes, and 50 min for each class, and the inference of the research results may be limited by time.

*5.4. Practice and Research Implications*

From the study, we found that the AIOT curriculum can be effectively implemented in the high school information curriculum, and the best teaching model is the theory course combined with the physical operation. The theory course can be taught by teachers in the classroom and assisted by the teaching platform, but not every student can easily get the hang of it. During the teaching experiment, the AIOT physical operation was demonstrated by the researcher in the classroom and then handed over to the students to complete in groups. The students were assembled while the teaching team observed or guided the group with conditions. The researcher reflected on the experimental teaching of AIOT physical operation, and the implementation and enlightenment of the operation course are as follows: before the class, a video of AIOT physical operation was taken and placed on the teaching platform for students to use the time for pre-study. During the class, the

teacher demonstrated each operation process one by one, and the students operated and completed the assembly in groups, while the teacher took turns to observe or instruct each group. After the class, students could watch the AIOT video repeatedly to familiarize themselves with the operation steps. By using the video and review before, during and after the class, students could effectively improve their learning outcomes of AIOT theory and practical courses.

**Author Contributions:** Research planning, execution and writing, C.-C.T.; Literature collection and data analysis, Y.-M.C.; Logistics support and co-authoring, Y.-S.T.; Research integration and supervision of implementation, S.-J.L. All authors have read and agreed to the published version of the manuscript.

**Funding:** This research received no external funding.

**Informed Consent Statement:** Informed consent was obtained from all subjects involved in the study. All the participants have been informed about the aim of this study and of the use of data collected.

**Data Availability Statement:** Data sharing is not applicable to this article.

**Conflicts of Interest:** The authors declare no conflict of in terest. The funders had no role in the design of the study; in the collection, analyses, or interpretation of data; in the writing of the manuscript, or in the decision to publish the results.

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
