# Peer review of "Impacts of AIOT Implementation Course on the Learning Outcomes of Senior High School Students"

_education, doi:10.3390/educsci11020082_

Round 1

Reviewer 1 Report

The current study is interesting and generally well-documented. I would like to suggest some modifications, which are as follows:

  1. Abstract is too long and should not exceed 250-300 words (normally).
  2. The second paragraph of 1.1. does not have any reference. Please provide some.
  3. I would prefer to be followed the APA style in the Research method. For instance, I did not find anything regarding participants’ background etc. but only the analysis of questionnaires.
  4. Implications for practice and research should be provided.

Author Response

Thank you for your valuable comments during your busy schedule.
1. The abstract has been revised to about 300 words.
2. The second paragraph of 1.1 has been added to the literature.
3. The background of the research subject has been briefly described in Section 3 and Subsection 3.5.
4. Section 5.4 has been added as the inspiration and practice of this study.

Reviewer 2 Report

This work presents a very interesting study about and some representative results. However, some aspects need to be improved.

The paper is hard to understand and, in order to clarity, it really needs a deep reorganization regarding both text structure and information presented in tables, figures and comments.

There is an important and general problem with the acronyms. For instance, the term AIOT included in the tittle is not defined until line 466. Every term should be defined the first time it appears. Another example is the term TAM, included but not defined in the abstract. I would suggest using Technology Acceptance Model (TAM) instead. Moreover, SPOC-AIOT is used in lines 13, 84, 388 but it is not defined until line 465-466 and it is done partially.

Some of the paper contents need to be explain with more detail. Some others, like information presented in Table 1, are not relevant for the conducted study. I would suggest removing this table. If not, please consider another way of presenting information that does not involve crossing out the content. Something similar happens with Figure 6, I do not see the need to include this information.

The most difficult part to understand is the information included in tables. In my opinion, the included tables should be divided and redesigned to be more comprehensible and understandable. Moreover, the information regarding their contents should be described in more detail.

Sections and subsections should be introduced with more detail. For instance, there is no introduction to section 3 and just one single sentence was used to introduce section 3.1.

There are too many important and different aspects included in Section 3. In my opinion, it should be divided into three different sections: the dimensions analyzed, the course organization and the research model hypothesis used. Moreover, there are problems with the numeration included in subsubsections generates a lot of confusion. I am talking about definitions included in subsubsection 3.2.1: 1. Exploration, 2. Definition, … and in section 3.2.2: 1. Exploration, 2. Definition, …. I think subdivision would help to clarify these problems. Moreover, in line 338 there are two numbers “1”.

Another important key to solve is that the dimensions of SPOC-AIOT Scale included in table 4 are not described anywhere. Moreover, the justification for excluding some of these indicators should be more detailed.

There is a general lack of uniformity and consistency between the different figures. Some of them presents a very huge letter size (Figure 1 and Figure 11) while others (Figure 2 and Figure 4) present a much smaller size. Moreover, there is a problem with Figure 7, it does not properly fit in the page. In addition, the quality of the images is very poor, especially of figures 7, 8 and 9 which are the most important for the presented work.

I would also suggest not starting a sentence with a reference number (lines 217, 281, 305) because it leads to misunderstanding.

The questionnaire used is not included, could it be included?

Despite these comments, I would like to congratulate the authors for the performed study and encourage them to present a new version of the paper taking the suggestions discussed.

Author Response

1. The abbreviations have been revised according to the comments of the committee members.
2. Table 1 and Figure 6 have been deleted, and all the figures in this study have been redrawn for consistency.
3. The introduction of each section and sub-section has been revised according to members' comments.
4. Section 3.2.1 and 3.2.2 were applied to curriculum design (3.2.1) and teaching practice (3.2.2) according to the 4 stages (Discover-Define-Develop-Deliver) of the 4D double diamond design model. The definitions of each stage are different as shown in Figure 1.
5. The questionnaire is provided in the Annex.
6. Since this study is an experimental study with a small sample (36 students), SmartPLS was used to analyze the results and the tables presented are the output of the software.The valuable comments of the members will be the goal of the researcher's next submission.

Round 2

Reviewer 1 Report

I propose the acceptance of this study!

Author Response

Thanks to the reviewer for their valuable comments and sample references, I have benefited a lot.

Reviewer 2 Report

Authors have made an important effort to clarify the content and have improved the writing style, however some of my recommendations have not been addressed.

Information presented in tables is still hard to understand. As I mention on my first review, the included tables should be divided and redesigned to be more comprehensible and understandable. For instance, I 've attached an alternative presentation for Table 2, including a separation between the different items would help to a better understanding and would help to realize some missing information like referenced as ????.

In the other hand, some tables do not have a blank space before the next time and many of them are split in different pages.

Moreover, there are still confusion with the numeration included in subsubsections. Even though authors have made some changes, there are still some problems. For instance, in line 629 starts a numbered block which is hardly impossible to kwon what section belongs and which subject describes. In addition, considering the numeration used in section 3.2.1, in line 423 “1. stage 1.Exploration: Arouse Interest” should be “1. Stage 1. Exploration: Arouse Interest”. The same problem (missing space) occurs with stages 2 to 4.

Finally, I would suggest making some changes in the introduction. The last sentence does not seem to be a conclusion. I do not know whether because it should appear before the conclusions or because it should be rewritten. I would also suggest replacing the use of the ":" by a descriptive phrase.

Author Response

Thanks to the reviewer for their valuable comments and sample references, I have benefited a lot.

  1. The abstract section has been reworked
  2. The table part of the manuscript has been redesigned